# How to lead health care workers during unprecedented crises: A qualitative study of the COVID-19 pandemic in Connecticut, USA

**Oluwatosin O. Adeyemo**[1]*, **Stephanie Tu**[2], **Danya Keene**[3]

**1** Department of Obstetrics, Gynecology and Reproductive Sciences, Yale School of Medicine, New Haven, Connecticut, United States of America, **2** Yale School of Medicine, New Haven, Connecticut, United States of America, **3** Department of Social and Behavioral Sciences, Yale School of Public Health, New Haven, Connecticut, United States of America

* Oluwatosin.adeyemo@yale.edu

**Data Availability Statement:** All relevant data are within the manuscript of this qualitative study. Full transcripts/data cannot be shared publicly because of the sensitive nature of the qualitative data we

## Abstract

Health Care Leaders (HCLs) faced unprecedented challenges during the initial phases of the COVID-19 pandemic. Leaders played an important role in shaping the experiences of Health Care Workers (HCWs) during this time. However, research is needed on how HCWs experienced and characterized HCLs' response and support. The aim of our study was to examine HCWs' experiences with leadership and to identify aspects of HCLs' response that were effective in supporting HCWs in their roles during the early phases of the pandemic. This was a qualitative study based on open-ended semi-structured interviews conducted (June 1- July 18, 2020) with frontline HCWs during the first wave of the COVID-19 pandemic in Connecticut, USA. Participants (N = 45) included physicians, nurses, respiratory therapists and patient care assistants who worked in inpatient and outpatient settings in various specialties, roles and 3 health systems across Connecticut, USA during the COVID-19 pandemic. Participants were offered a $25 gift card as an incentive for participation. We used inductive techniques derived from grounded theory to develop themes. We identified 6 main themes related to leadership response and support of HCWs during the pandemic namely: 1) Effective communication and transparency; 2) Prioritizing their health and safety; 3) Employee scheduling considerations: autonomy, assignment support and respite; 4) Appreciation- financial and nonfinancial; 5) Showing up and listening and 6) Stepping up with resources. Our findings can inform leadership responses to future pandemics and other unanticipated crises leading to strengthening of the health care system as a whole.

## Introduction

Leadership is critical in moments of crises. Health care systems globally faced an unprecedented crisis during the initial phases of the COVID-19 pandemic. Health care systems in traditionally high resource settings like the United States faced unusually high levels of resource limitations including scarcity of personal protective and other medical equipment required to

collected. Study participants shared very personal and sensitive information about their employers, co-workers, patients and personal lives that could easily identify them, their work associates and hospital systems if we make the full transcripts available and could have negative consequences on study participants. We do provide relevant parts of the transcripts in the manuscript. Our IRB consent for the study guaranteed confidentiality and anonymity to the study participants. Data are available from Yale University Ethics Committee (contact via irb.support@yale.edu) for researchers who meet the criteria for access to confidential data. Please feel free to let us know if there are any other questions or concerns.

**Funding:** The authors received no specific funding for this work.

**Competing interests:** The authors have declared that no competing interest exist.

keep patients and health care workers safe and healthy [1–3]. The crises caused by the pandemic undoubtedly placed a huge burden on health care workers (HCWs) as well as health care leaders (HCLs) tasked with the responsibility of supporting HCWs.

A report published in JAMA shed light on requests HCWs made of HCLs at the beginning of the pandemic such as to be heard, protected, prepared, supported and cared for [4]. However, the perspectives in this article were elicited during the first week of the pandemic and thus unable to capture lived experiences during this rapidly evolving time. Similarly, there have been a number of editorial papers published on the role of HCLs in supporting HCWs during the COVID-19 pandemic however these papers were not based on experiences of HCWs [5–10]. Additional recent studies published highlight the experiences and stressors HCWs faced during the initial phases of the pandemic, however more research is needed [11, 12].

Specifically, research is needed to understand the role of leadership in providing support to HCWs during the pandemic. A qualitative study in New York, a state with record-high COVID-19 cases during the first wave of the pandemic, focused on the lived experiences of home health care workers during the first wave of the pandemic in New York and highlighted the challenge of the varying levels of institutional support they received [13]. However, given the different set of challenges faced by HCWs in more acute patient care settings like hospital and outpatient clinics, there is a need to further explore the experiences of HCWs within these settings.

The goal of our study was to explore the experiences of a broad range of HCWs who provided patient care during the first wave of the COVID-19 pandemic in Connecticut. Connecticut was one of the states in the northeast of the United States with high cases and hospitalizations during the first wave of the COVID-19 pandemic that peaked in April, 2020 making the experiences of HCWs in Connecticut relevant with regards to understanding responses to unprecedented crises [14]. In this qualitative analysis, we examined the experiences of HCWs to identify aspects of HCL response that were effective in supporting HCWs in their roles. HCL in this study is defined broadly as anyone with a supervisory capacity clinically or administratively including senior residents, attendings, heads of department or managers. Our findings can inform the quality and effectiveness of HCLs' response and support of HCWs in other related health care crises.

## Materials and methods

We conducted a qualitative study utilizing in-depth, open-ended interviews with HCWs who provided care to patients during the initial wave of the COVID-19 pandemic. Participants were recruited through emails sent through hospital and university listservs of health care systems (HCSs) in Connecticut, USA stating the goals of the study. The study was given exempt status by the Yale University institutional review board. Participants provided verbal informed consent. The first author O.A., is a practicing obstetrician and gynecologist with training in qualitative methodology, S.T. is a medical student and D.K. is a public health professor and researcher with extensive experience with qualitative methodology.

### Study recruitment and sample

We contacted via email, leaders at various HCSs across the state of Connecticut, USA including hospital department chairs, division chiefs, residency program directors and nursing leaders. We asked the leaders to forward our recruitment email to their staff. The recruitment email stated that we were inviting HCWs working in Connecticut during the COVID-19 pandemic to participate in an interview to learn about their experiences during the pandemic. All

potential participants were further screened for eligibility via a short email survey (see S1 File). Individuals were eligible to participate if they were HCWs including resident and attending physicians, respiratory therapists, nurses, patient care assistants (PCA) and certified nurse midwives; worked in Connecticut as a HCW since March 2020 in the inpatient hospital, out-patient clinic or both settings and had in-person clinical encounters with patients during the pandemic.

A total of 98 HCWs who responded to the initial inquiry. Of these, 38 did not complete the screening questions sent via email. Of the 60 individuals who responded and completed the screening questions sent via email, 2 were ineligible to participate. We interviewed 45 individuals out of the 58 eligible individuals. We purposefully sampled these participants to maximize the number of HCWs who had cared for a relatively high number of patients with COVID-19 diagnosis in hospital in-patient settings who were critically ill and to balance the participants with respect to their specialty and role. We performed concurrent analysis of our interview transcripts and stopped further recruitment and interviewing when we reached thematic saturation. Study participants worked in varying roles and specialties across 3 major HCSs in Connecticut (Table 1).

## Data collection

We performed one- on–one interviews virtually from June 1st to July 18th, 2020. All three authors (O.A., S.T. and D.K.) developed the interview guide through a process of team discussion that was informed by one author's experience (O.A.) working in the hospital as a physician early in the pandemic and our review of emerging popular and academic literature about HCW experiences. The interview questions focused on broad experiences of HCWs during the pandemic such as their clinical experiences, experiences with supervisors and co-workers, resource limitations, as well as their experiences at home and with the non- medical community (see S2 File). Specifically, with regards to HCLs, we asked participants open-ended questions that directly explored their perception of leadership's response and support during the pandemic. These questions include 1) Can you tell me a little bit about how your hospital/ department has responded to COVID-related risks? 2) Is there anything else your institution could have done to support you? Given the semi-structured nature of our questions, we left it up to the participant to choose what aspect of hospital, departmental or institutional leadership they wished to focus on and what "support" meant to them. We also further explored the role of HCLs when participants mentioned the role of leadership in other experiences they had such as availability of personal protective equipment (PPE), impact on their scope of work, etc. We collected self- reported demographic data such as age, gender, race/ethnicity, marital status, number of children, role, specialty and number of patients with COVID-19 cared for. All 3 authors conducted interviews virtually via the online secure version of the Zoom software and the interviews lasted 47 minutes on average. Participants received $25 gift card as a token of appreciation. The interviews were audio recorded and transcribed with an online transcription tool (Trint), after which, the transcripts were edited by a research assistant for accuracy.

## Data analysis

Our analysis in this paper focuses on HCWs experiences and perceptions of HCLs' response and support at their institutions. We used inductive techniques derived from grounded theory to develop themes [15]. Interviewers wrote reflexive memos after each interview and met regularly to discuss emergent concepts. Additionally, we used an iterative and multi-stages coding process that draws on grounded theory approaches [15]. First two of the authors (O.A. and S. T.) reviewed the transcripts and performed open coding of 9 transcripts independently to

**Table 1. Characteristics of study participants.**

|  | Number (Percentage) Total N = 45 |
|---|---|
| **Specialty** |  |
| Internal Medicine | 16(35.6%) |
| *Internal Medicine Subspecialties | 8(17.8%) |
| Respiratory Therapy | 5(11.1%) |
| Obstetrics and Gynecology | 10(22.2%) |
| †Other | 6(13.3%) |
| **Role** |  |
| Attending Physician | 15(33.3%) |
| Resident Physician | 19(42.2%) |
| Respiratory Therapist | 5(11.1%) |
| Registered Nurse | 3(6.6%) |
| PCA/Tech | 2(4.4%) |
| Certified Nurse Midwife | 1(2.2%) |
| **Practice Setting** |  |
| Inpatient only | 31(68.8%) |
| Outpatient only | 2 (4.4%) |
| Inpatient and outpatient | 12(26.6%) |
| **Number of COVID patients cared for** |  |
| 0 | 4 (8.9%) |
| 1 to 5 | 7 (15.5%) |
| 11–20 | 4 (8.9%) |
| >20 | 30 (66.6%) |
| **Age** |  |
| 25–29 | 12(27.0%) |
| 30–34 | 13 (29.0%) |
| 35–39 | 10 (22.0%) |
| 40–44 | 4 (8.9%) |
| 45–49 | 1 (2.2%) |
| 50–54 | 3 (6.6%) |
| 55–59 | 2 (4.4%) |
| **Gender** |  |
| Male | 15 (33.0%) |
| Female | 30 (67.0%) |
| **Race** |  |
| Caucasian | 33(73.3%) |
| Asian | 6(13.3%) |
| ‡Other | 6(13.3%) |
| **Marital Status** |  |
| Not married | 23(51.0%) |
| Married | 22(49.0%) |
| **Children** |  |
| Yes | 20(44.4%) |
| No | 25(55.5%) |

*includes pulmonary critical care, palliative care, hepatology, geriatrics, infectious disease, hematology/oncology.

†Includes emergency medicine, pediatrics, anesthesiology, PCA/Tech.

‡includes Black, Middle Eastern, Hispanic.

identify emergent concepts. These concepts were refined through ongoing discussion among all 3 authors and subsequent rounds of open-coding to develop a focused codebook. Two of the authors (O.A. and S.T.) then used this codebook to code the remaining 36 transcripts. All three authors met regularly to discuss code application, making small adjustments where necessary. To consolidate the leadership themes, one of the authors (O.A.), reviewed all excerpts pertaining to HCWs experiences with or perception of HCLs at all levels of leadership and also reviewed full transcripts to contextualize these excerpts. The first author also wrote integrative memos to develop relationships between codes and concepts and discussed these memos with other team members (D.K. and S.T.).

We used the qualitative analysis software Dedoose (Version 8.3.35, web application for managing, analyzing, and presenting qualitative and mixed method research data, 2020) to facilitate analysis.

## Results

We identified 6 recurring themes related to what HCWs valued about their leadership's response and support during the first wave of the pandemic. These themes include: 1) Effective communication and transparency; 2) Prioritizing their health and safety; 3) Employee scheduling considerations: autonomy, assignment support and respite; 4) Appreciation- financial and nonfinancial; 5) Showing up and listening; 6) Stepping up with resources.

### Effective communication and transparency

Communication and transparency played important roles in how HCWs judged the response of HCLs. Frequent communication provided a sense of control for many participants and helped them manage their anxiety and fear. For example, one pediatric attending describes how frequent communication about the availability of PPE in the hospital helped allay anxiety related to PPE shortages. They noted:

> "…. And I think the components of hospital leadership, making sure there is enough PPE, and communicating with us daily about the status of the PPE was really helpful. And eased some of the anxiety around running out and why we weren't stressed out about it…"

(Participant 22)

Despite the benefits of frequent communication, participants described a threshold where increased frequency became ineffective. For example, participants described frustration with receiving multiple versions of policy changes within a short period of time or receiving information prematurely. As one OBGYN resident, describing the mental burden associated with superfluous communication, noted,

> "…I think there is a huge component of notification or email fatigue, so to speak. You know, we got, especially in the beginning, this daily email from our program director, a daily e-mail from the hospital president, the daily e-mail from God knows who, there was at least three or four daily e-mails, all of which had attached six or seven new policies or modifications to some random policy, half of which affected you, half of which didn't…."

(Participant 27).

HCWs also appreciated leaders who were honest with them about threats and concerns. In contrast, lack of transparency contributed to mistrust, as was evident in discussions around

PPE. Some participants described lack of transparency around PPE scarcity as an insult to their intelligence that led to erosion of trust. As one attending physician noted:

". . .And then we were being told things like that we all know actually are not appropriate for our general PPE, like using your mask over and over. . .So they were it almost felt like gaslighting a little bit. I think that part, lack of transparency, could have been improved upon . . .there's a sense of trust that's sort of lost when you're telling people things that completely go against their knowledge. . ."

(Participant 43)

Perceived transparency from HCLs created a more trusting environment for HCWs, which in turn had a positive impact on their work. For example, one OBGYN attending described how transparency by leadership gave them mental bandwidth to focus on caring for patients. They noted,

". . .Because I felt like they were being so transparent with what was going on. I didn't feel compelled to kind of question it. And, you know, maybe it's not that I wouldn't question it. . .But their transparency with these daily emails, daily stats, and daily updates, to me was very reassuring. They're not lying to us. . ."

(Participant 33)

## Prioritizing their health and safety

HCWs appreciated leaders who prioritized their health and safety in the midst of PPE and testing scarcities that were common in the early phases of the pandemic. In particular, HCWs appreciated clear guidelines on managing situations that created conflict between their safety and their clinical responsibilities. For example, one internal medicine (IM) resident described how their leadership's clear direction helped them navigate the scarcity of PPE. They noted,

". . .They made sure to give us rules about how often to reuse and recycle PPE. That did kind of change from day to day to week to week, which could be confusing. But overall, I think those rules made it so that we always had the protection that we needed."

(Participant 4)

Some HCWs described scenarios where they struggled with the dilemma of whether to prioritize their own health and safety versus their patient's in the setting of PPE scarcity. Other HCWs, particularly trainees, expressed how clear guidelines from leaders during those periods of PPE scarcity were useful in navigating these tensions, as described by this IM resident:

"This patient is coding and there is no choice. And I just kind of went in there and did my job. But moving forward, it was very clear, actually, that we would even get weekly e-mails from our chiefs and kind of confirmed by our program director. By no means are you expected to step into a room without the proper full PPE and that's 100 percent supported, even if that means the patient is coded and the code gets delayed in being started. You just have to have the proper PPE before going in. . ."

(Participant 11)

However, guidelines that prioritized HCW safety did not eliminate some of the tensions with how to prioritize personal safety over patient care or tensions around disproportionate exposure to the virus among HCWs. For example, a pulmonary critical care attending described situations where some HCWs did not go into rooms to evaluate patients when needed even when they had PPE which created tensions between nurses (who typically spent more time at the patient's bedside) and other providers.

"...either the APP [advanced practice provider] or the house staff weren't going into patient rooms and the nurses kind of felt abandoned and alone in the rooms. Right, where patients were either coding or very sick and everybody standing outside the door and not going in the room because they're afraid to go in the room. Despite having their PPE..."

(Participant 40)

The above quote also suggests that factors beyond availability of PPE contributed to the feeling of safety by HCWs during the early phases of the pandemic. Uncertainty about the infectivity of the disease, mode of transmission and effectiveness of PPE likely contributed to the complexity of how HCWs perceived safety as the following quote from an OBGYN resident illustrates:

"... And then in the very beginning of the pandemic, the unknown was obviously very anxiety provoking. Was the PPE we were wearing good enough?... Could I contract the virus and bring it home to my wife...?"

Participant 32

Notably HCWs developed more confidence in the PPE's ability to be protective as time went by and more data points emerged as the following quote from a respiratory therapist illustrates:

"So overall, it was, I think, one of the most stressful periods of my life. Just because you don't know if you're going to get this, if you're going to bring it home. And of course, you're protecting yourself as much as possible but they didn't really know a lot of information in the beginning like how it was transferred. You know, I guess after the first month or so, it kind of started to become a little bit more normal. And then when you see all these therapist and nurses are directly seeing these patients and they're not getting sick, whatever we're doing must be the right way..."

Participant 18

## Employee scheduling considerations: Autonomy, assignment support and respite

Several participants discussed work scheduling as an important aspect of pandemic leadership response. Many HCWs volunteered to work during the pandemic and were proud that they could provide valuable expertise. However, other HCWs described a lack of choice. For example, one PCA who worked in an outpatient setting but was re-assigned to work in an inpatient setting. They described the pressure they felt to preserve their reputation by working as well as concern about their job security if they chose not to work. They explained,

"...But we didn't have a choice. Yeah, I felt like, you know, if you were to decline being sent into the hospital, you would have been looked very poorly upon. Like you would have been look down upon. And I don't know, you might not have had a job after..."

(Participant 44)

An attending physician who felt they weren't given a choice to work in the intensive care unit (ICU) suggested that leaders should consider giving HCW's an opportunity to volunteer for these positions. They explained that many HCWs would volunteer and that the opportunity to make this choice on their own would have been valuable:

"So, you know, I feel like the way I would have approached it has been like, hey, you know, just email everybody. And, you know, quite honestly. Have they done it that way, I probably would have still volunteered, but it would have just been a more palatable way of being assigned in the ICU or step down versus it being like, oh, hey, this is the expectation and you kind of don't have a choice...and like, I feel like they would have had an adequate number of volunteers by doing that and, you know, obviously, if they didn't, then they could have gone to a more paternalistic approach, you know . . ."

(Participant 24)

The tension between volunteering to work versus being forced to work was likely nuanced by the uncertainty about COVID-19 disease and risks of being infected during the early days of the pandemic. The impact of this uncertain risk was noted by another attending physician who stated,

"...You know the argument being made that this is what you signed up for so you should just do it and stop complaining about it. Because some people, they didn't want to do. Some people were reasonably uncomfortable to work in such settings and I could understand that. But I heard the argument from some peers again and again that, you know, it's your job, you need to just do it...And I thought those were inappropriate comments because who signed up to care for a disease that we don't know much about, without good equipment? I don't think any of us like when we sign up for medical school to sign up for that specifically. ..."

(Participant 25)

Although many HCWs reported good adaptation to changes in their clinical roles and assignment without negative impact on patient care, there were HCWs who reported observing negative patient outcomes associated with staffing changes. One respiratory therapist noted,

"I think people died not as a result of COVID, you know, having it, I believe that they some of them died because we didn't have enough staff to properly evaluate every little nuance, every little thing that that patient needed was not addressed..."

(Participant 45)

Similarly, a pulmonary critical care attending expressed that adverse outcomes were a result of HCWs not working with patient populations they are familiar with:

". . .I guess it was like desperate times call for desperate measures, and I mean, and so I honestly think some of our, not our adverse outcomes, but things didn't go as well as they could if we had had our experienced or experienced ICU nurses, because, you know, they don't do things the way our nurses are trained, like, you know, they wouldn't titrate the drip down so people would get more meds than they really needed or, you know, they just weren't comfortable with things. So, they didn't want to change anything."

(Participant 40)

Therefore, an important consideration for HCLs was how to provide support for HCWs re-assigned to work in unfamiliar settings. In addition to the protocols and guidelines that were frequently distributed, HCWs working with unfamiliar patient populations described the benefit of being scheduled with someone else with more expertise to review patient care plans with. As noted by an IM attending who was re-assigned to care for patient populations they had not cared for since many years ago in residency:

". . . there were two attendings assigned to the unit and the other attending was actually trained in [medicine subspecialty]. So, I talked to him and he told me, he's like oh, here's what you can do. . ."

(Participant 31)

HCWs who worked during the pandemic also valued built-in periods of rest in their schedule. They appreciated having a few days off to get away from the hospital, which contributed to their sense of wellness. As one IM resident noted,

". . .And so, they've really been very deliberate to make our schedules very humane. So, we have you know, we'll work for like a chunk of time and then we'll have a chunk of time off or on backup. That sort of alternation and being able to count on some unwinding times has been built into the schedule, which has been really wonderful. . ."

(Participant 3)

## Appreciation- financial and non-financial

Several participants discussed appreciation by HCLs as a crucial part of feeling supported during the pandemic. Many HCWs viewed extra financial compensation as a way for HCLs to truly express appreciation for their extra work and harmful exposure. There was dissatisfaction when leaders ignored the issue of extra financial compensation as exemplified by a quote from an IM attending,

". . . I think the management, which is really upper level, whoever would be totally, like, honest and real and just say, like, hey, guys like, you know, elective surgeries are canceled and like this is canceled and outpatient ambulatory clinics are closed. And, you know, and like, we're losing one and a half million dollars a day at this point on average. . .So, you know, we are so thankful for you, for your service. And we will, like, offer hazard pay if it's like possible, you know, with time or something like that, like just instead of just like avoiding the topic. . .".

(Participant 24)

There was no clear consensus among the HCWs in this study on how much extra financial compensation was appropriate. Some HCWs were glad to get any amount of extra compensation while others felt the extra financial compensation should vary based on level of exposure or extra work done. One PCA explained that the one-time bonus or hazard pay they received was not sufficient compensation for the risks associated with repeated exposure to the virus at work and would have preferred continued hazard pay:

"...I would say our hospital gave us this one time pay so that individuals wouldn't feel, not unappreciated, but more so undervalued. Like they were, I think, especially myself, like if you're being exposed to COVID every single shift and your employer says well, we'll pay you this bonus one time. But please continue to expose yourself every shift following this. And we're gonna continue to pay you the same amount regardless of the presence of the pandemic. It's kind of a bit insulting because it's almost as if your health and your livelihood aren't being considered in this process..."

(Participant 39)

In contrast, other participants described how a one-time extra financial compensation made them feel supported especially during a period where many other employers were losing their jobs. As an OBGYN attending noted,

"And there were simple things as well that went a long way, like we were constantly being given messages of support. Early on, we were given a financial bursary, which was very surprising. But it came at a perfect moment because we were hearing a lot of hospitals were laying off staff and nurses were losing their jobs...That was a fantastic measure of support..."

(Participant 34)

Despite the value placed on financial compensation as a form of appreciation, many HCWs also placed high value on simple but profound gestures of appreciation particularly public acknowledgment of their efforts. A PCA who was deployed to work in the inpatient setting described how at the end of their assignment, their manager sent a letter to their head of department commending them for their work during the pandemic. She explained,

"...Oh, I'm so proud, very happy and just very appreciative because that doesn't happen in nursing..."

(Participant 44)

### Showing up and listening

HCWs wanted the physical presence of HCLs, when possible, to provide support and guidance. One respiratory therapist described how the physical presence of their direct supervisor to troubleshoot challenges provided a sense of comfort particularly if it was a familiar face. They explained,

"...And those situations just having more bodies around, like having the supervisors, you know, show up on the floor or that really we rely on our charge therapists a lot for those situations and just having that like you, knowing that somebody is there is definitely more

comforting than anything because you want to be able to call someone to be like, you know, I need help. . .”

(Participant 9)

In contrast, the physical absence of a leader could lead to discouragement as well as a perception that the leader was ineffective. For example, one attending physician explained how physical absence prevented their departmental leader from responding to the rapid changes occurring early in the pandemic. They noted,

"Unfortunately, our department head, [they] stopped really coming into work. . .So all of [their]meetings are being run from home and [they] didn't really have [their] finger on the pulse anymore as to what was actually taking place in hospital. . . I felt quite discouraged by that. . .”

(Participant 34)

HCWs appreciated supervisors who created platforms for their concerns to be heard and listened to their feedback on how to manage the crises. One respiratory therapist reported feeling supported by her manager because the manager was attentive to her concerns, noting:

". . . There was a couple of times I go to her office and just. . . And, you know, any concern I had, it would be addressed and it would be brought directly to who's above them. I would get an email back. . . "

(Participant 18)

Given that large gatherings were prohibited, HCWs appreciated the creation of alternate forms of community engagement and listening such as virtual town halls. As noted by one IM resident,

". . . Town halls are sort of led by my program director. A lot of involvement from the chief residents. . .It's like a partial venting space to people . . .I think the town halls have been really useful in making sure that we feel heard and at least try to get some of our questions answered. . .”

(Participant 3)

## Stepping up with resources

HCWs appreciated leaders who were resourceful and who made visible efforts to tackle the unprecedent challenges. Many HCWs commented on the resourcefulness of their leaders during the initial phases of the pandemic such as creating telehealth infrastructure, establishing new patient care units, sourcing for PPE, creating new clinical protocols and guidelines amongst other things. HCWs took note of the speed with which leaders implemented these changes and it served as a metric for judging the quality of leadership's response. One IM resident viewed the leadership's response as positive because they were swift to act in providing infrastructure to facilitate care for the high volume of patients admitted COVID-19. They noted,

"I felt like the [institution] had a pretty good response up front. They just ended up converting one entire floor into at least the floor patients for COVID. And I feel like they were pretty quick about things"

(Participant 13)

Similarly, HCWs appreciated the quick response of HCLs in providing technical support as new patient care guidelines were rolled out as noted by a certified nurse midwife:

I was impressed by the hospital's quick response, especially with training people. . ., the people who are like standing at the donning and doffing station, who are telling you what orders to put things on and what orders to take things off. . .

(Participant 29)

Apart from resourcefulness for patient care, HCWs appreciated leaders who stepped up to address their personal needs including their emotional and psychological wellbeing. HCWs appreciated tangible provision of resources for their personal needs as such provision of alternative accommodation when needed as this quote from an OBGYN resident describes:

I mean, I think they really thought about a lot of things, they thought about housing for us . . .So they, like, thought a lot about ways to help people, especially COVID positive people. . ."

(Participant 3)

Or provision of food like one IM resident noted:

And then from a program and hospital standpoint, our hospital did an awesome job with feeding us, I should say. We had we had meals for, all three meals the day, sometimes too much food, sad to say . . .

(Participant 17)

As expected, childcare was a major source of concern for healthcare workers with children during the pandemic and HCWs appreciated it when leaders cared about this issue even if the solution was a work in progress as this IM attending noted:

Yeah, I mean, I think overall I felt very supported, you know. And listen, if I had, for this whole issue of child care and what if we got sick, I really felt like the hospital was trying to do the best they could to figure it out.

(Participant 31).

## Discussion

Health care leaders and organizations play a critical role in supporting HCWs particularly during moments of crises [16]. A framework for best leadership practices during public health crises includes accepting and involving stakeholders as legitimate partners, listening to people, being truthful, honest, frank and open, coordinating and communicating clearly and with compassion [17].

Our study identified 6 themes that can be used to assess HCL's response and support of HCWs during the COVID-19 pandemic. We highlight what it meant to HCWs to feel supported by their leaders during the initial phases of the pandemic such as the importance of transparency and honesty in leadership's handling of the unprecedented challenges the COVID-19 pandemic posed. Despite the importance of communication during a crisis, we note the delicate balance HCLs needed to strike in communication frequency and volume in order to keep HCWs informed but yet not create anxiety and confusion. This need for balance in an age of rapid electronic communication was echoed by research in other settings [18].

We also highlight the balance between HCW autonomy and paternalism in work assignment by HCLs. Lack of control was an important factor that contributed to distress among HCWs during the pandemic [12], and our study highlights how HCLs can ease the burden for HCWs with regards to scheduling by providing HCWs autonomy and respite.

Our study further explores the different expectations among HCWs about extra financial compensation that leaders need to consider including not avoiding conversations about finances no matter how difficult and sensitive such conversations could be.

Furthermore, we highlight the important role that physical presence and visible efforts of HCLs played in boosting the positive appraisal of leadership among HCWs even in the presence of scarcity and suboptimal working conditions.

Results from our study echo findings from a study performed at the beginning of the beginning of the pandemic about the requests from HCWs to their organization to be heard, protected, prepared, supported and cared for [4]. Our study builds up on this prior work by providing empirical evidence of how HCWs experiences leadership during the first wave of the pandemic.

Our study also provides empirical evidence that supports theoretical approaches to leadership during emergency situations and crises such as the importance of transparency, physical presence, effective communication, addressing basic needs and provision of support including mental health support [5–9]. In addition, our findings support existing literature on the responsibility of leaders to provide psychological safety to team members by acknowledging mistakes made by leadership and to create a safe space for dissenting views to be expressed by team members [19]. Furthermore, our findings on how lack of transparency and poor communication by HCLs could create mistrust during a crises is consistent with other qualitative work conducted among clinicians and clinician leaders in the United States during the COVID-19 pandemic [20].

HCWs in our study described the impact that multiple levels of leadership had on them starting from those with direct oversight such as chief residents (who were involved in scheduling, communication and listening forums), attending physicians, unit managers; those with higher order oversight such as residency program directors, section chiefs, departmental chairs and to those with more centralized leadership roles such as hospital chief executive officers.

Lastly, narratives in our study about the challenges HCWs faced with job re-assignments and limited resources further contributes evidence to support the call for HCLs to provide HCWs with emotional and psychological support, now and in the future, as they process events and stressors they faced during the pandemic [21–23].

A strength of this study is that it provides perspectives based on lived experiences of a variety of HCWs at various HCSs across the state of Connecticut. We also include the experiences of less publicized HCWs like RTs and PCAs. Although the proportion of non-physician participants in our study was limited, their narratives provide a more comprehensive view of experiences of HCWs during the initial phases of the pandemic. In addition, the qualitative nature of our study allows us to identify nuances and tensions in how HCWs wanted to be supported during the pandemic.

A limitation of the study is that the participants were restricted to HCWs that practiced in Connecticut during the pandemic. However, many of the themes we identified mirror themes outlined in other settings and by health crises experts [5–13, 24, 25]. Another limitation were the challenges with recruiting registered nurses to participate in our study. There is a need for more research that focuses on the unique experiences of nurses given that they spend more time at the bedside with patients compared to other providers and may have experiences unique challenges as a result of their patient facing roles [25].

Our hope is that lessons from this study, derived from lived experiences of HCWs, can be applied by HCLs at various levels and health care organization to effectively support HCWs in future pandemics and other related crises and in so doing, strengthen our health care system.

## Supporting information

**S1 File. Screener email.**
(DOCX)

**S2 File. Interview guide.**
(DOCX)

**S1 Checklist. COREQ (COnsolidated criteria for REporting Qualitative research) check-list.**
(PDF)

## Acknowledgments

The authors would like to thank Whitney Menary for her work with transcribing the interviews. We would also like to thank the participants of this study for sharing their experiences with us.

## Author Contributions

**Conceptualization:** Oluwatosin O. Adeyemo, Danya Keene.

**Data curation:** Oluwatosin O. Adeyemo, Stephanie Tu, Danya Keene.

**Formal analysis:** Oluwatosin O. Adeyemo, Stephanie Tu, Danya Keene.

**Methodology:** Oluwatosin O. Adeyemo, Stephanie Tu, Danya Keene.

**Writing – original draft:** Oluwatosin O. Adeyemo.

**Writing – review & editing:** Oluwatosin O. Adeyemo, Stephanie Tu, Danya Keene.

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
