## [Decision Letter · Decision Letter 0]

30 May 2021

PONE-D-21-11313

How to lead health care workers during unprecedented crises: A qualitative study of the COVID-19 pandemic in Connecticut, USA

PLOS ONE

Dear Dr. Onibokun,

Thank you for submitting your manuscript to PLOS ONE. After careful consideration, we feel that it has merit but does not fully meet PLOS ONE’s publication criteria as it currently stands. Therefore, we invite you to submit a revised version of the manuscript that addresses the points raised during the review process.

We look forward to receiving your revised manuscript.

Kind regards,

Tim Luckett

Academic Editor

PLOS ONE

Journal Requirements:

4. Please include a copy of the interview guide used in the study, in both the original language and English, as Supporting Information, or include a citation if it has been published previously.

Reviewers' comments:

Reviewer's Responses to Questions

**Comments to the Author**

1. Is the manuscript technically sound, and do the data support the conclusions?

Reviewer #1: Yes

Reviewer #2: Yes

2. Has the statistical analysis been performed appropriately and rigorously? 

Reviewer #1: Yes

Reviewer #2: Yes

3. Have the authors made all data underlying the findings in their manuscript fully available?

Reviewer #1: Yes

Reviewer #2: Yes

4. Is the manuscript presented in an intelligible fashion and written in standard English?

Reviewer #1: Yes

Reviewer #2: Yes

5. Review Comments to the Author

Reviewer #1: The authors report an inductive qualitative analysis based on semi-structured interviews with healthcare workers who practiced during the COVID pandemic in 3 healthcare institutions in Connecticut and were interviewed June-July, 2020. They focus their analysis on perceptions and experience with hospital leadership and describe 6 themes describing the importance of transparency, prioritizing safety, considering work scheduling, expressing appreciation, in-person communication, and providing resources. This topic is important and the real-world experience of clinicians, elicited in real-time offer empirical evidence to support emergency planning guidelines largely based on theory. The inductive qualitative approach is an optimal choice to support better understanding experiential phenomena during the pandemic because these experiences are unprecedented and existing theory insufficient. The authors are to be commended on completing a substantial amount of work, especially during a time that may have offered challenges for recruitment and analysis.

I do think that findings would be more valuable and novel with a bit more development if the data is sufficiently rich to support elaboration. The authors can also offer more detail in their description of methods. I have several specific recommendations below:

Introduction

• The authors could refer to several other recent works related to the experience of healthcare workers during the pandemic: Ohta et al. “overcoming challenges” JGIM 2020; Sterling et al. “experience of home health care workers” JAMA IM, 2020; Butler et al. “professional roles and relationships”, BMJ open, 2021 (disclosure -- this is work by my group); Rosenberg et al. “exploring the impact” J Pain Symptom Manage, 2021

Methods

• Can the authors clarify their approach to recruitment? How many HCWs responded to the initial inquiry? Were the 45 interviewees selected purposively? If so, what guided this sampling? Was recruitment informed by concurrent analysis? Did the analysis reach thematic saturation?

• Can the authors be more specific on which analytic tools from grounded theory methods they draw from? Can they elaborate on the step of “consolidating themes”?

• Can the authors include a sample interview guide and describe how this guide was developed? Specifically, it is not clear how participants were asked about their experiences with leadership and whether they shared the same understanding of leadership as the authors offer in the introduction. Most of the comments seem to relate to administration or institutional leadership rather than less formal leadership roles. Were the participants asked about their own experience as leaders? The perspective of people with leadership roles might offer a more multi-faceted understanding of the entire phenomenon of leadership during the pandemic from multiple stakeholder perspectives.

Results

• Overall, the findings are mostly descriptive and I wonder if these could be developed a bit more deeply to offer more novel insights if the data is sufficiently rich to support this. For example, (Theme 2, para 3) What is it about some situations that HCWs still wouldn’t enter rooms despite having PPE? (Theme 3, para 2) What is it about volunteerism (vs being made to work) that felt important? (Theme 5) What was it about leaders being in-person that HCWs appreciated?

• Most of the discussion seems to relate to administrative or institutional leadership more than broad conceptualization of leadership suggested by the authors in the introduction. Can the authors comment on this?

Discussion.

• The authors mention in the introduction that theoretical approaches to leadership during emergency conditions exist, but need to be informed by empirical work. Can they comment on how their work relates to the existing work that they reference?

• Paragraph 6. I am not sure how the findings support multiple levels of leadership. Can the authors elaborate?

• Discussion might be reinforced by the work of Tannenbaum on team dynamics and leadership (eg, Tannenbaum et al. Managing teamwork in the face of pandemic” BMJ Qual Saf, 2021)

Reviewer #2: I commend the authors for their work. This is informative and valuable information for hospital leadership who may face future healthcare crises. This manuscript adds to the growing literature of how hospital systems and healthcare leaders responded to the COVID-19 pandemic. Furthermore, this study add to the literature by exploring how healthcare workers experienced the pandemic as they were thrust into sometimes a chaotic situation of reassignments and uncertain risk. Overall, I think this is a worthy study for publication.

6. PLOS authors have the option to publish the peer review history of their article (what does this mean?). If published, this will include your full peer review and any attached files.

Reviewer #1: **Yes: **Catherine Butler

Reviewer #2: No

---

## [Author Response · Author response to Decision Letter 0]

17 Jul 2021

Academic Editors Comments and Responses

Response: Done

Response: Done. Please see details below

Response: Done

4. Please include a copy of the interview guide used in the study, in both the original language and English, as Supporting Information, or include a citation if it has been published previously. 

Response: Attached

Response: Also noted above

Given that our study is a qualitative study, we shared excerpts of the transcripts relevant to the study in the manuscript as recommended by the PLOS journal guidelines below. Based on the agreement made with study participants at time of consent and in order to protect their confidentiality and anonymity given the sensitive matters discussed it would not be appropriate to share the full transcript. Please let us know if our interpretation is not in-line with your journal guidelines below:

“For studies analyzing data collected as part of qualitative research, authors should make excerpts of the transcripts relevant to the study available in an appropriate data repository, within the paper, or upon request if they cannot be shared publicly. If even sharing excerpts would violate the agreement to which the participants consented, authors should explain this restriction and what data they are able to share in their Data Availability Statement”. https://journals.plos.org/plosone/s/data-availability

b) If there are no restrictions, please upload the minimal anonymized data set necessary to replicate your study findings as either Supporting Information files or to a stable, public repository and provide us with the relevant URLs, DOIs, or accession numbers. Please see http://www.bmj.com/content/340/bmj.c181.long for guidelines on how to de-identify and prepare clinical data for publication. For a list of acceptable repositories, please see http://journals.plos.org/plosone/s/data-availability#loc-recommended- repositories. 

Response: Please see above

Response: Done

Reviewer comments and responses

Introduction

The authors could refer to several other recent works related to the experience of healthcare workers during the pandemic: Ohta et al. “overcoming challenges” JGIM 2020; Sterling et al. “experience of home health care workers” JAMA IM, 2020; Butler et al. “professional roles and relationships”, BMJ open, 2021 (disclosure -- this is work by my group); Rosenberg et al. “exploring the impact” J Pain Symptom Manage, 2021 

Thank you for these additional references to make our introduction more robust. We included the Sterling et al, 2020 reference as stated below: 

“A qualitative study in New York, a state with record-high COVID-19 cases during the first wave of the pandemic, focused on the lived experiences of home health care workers during the first wave of the pandemic in New York and highlighted the challenge of the varying levels of institutional support they received [Sterling et al ,2020]. However, given the different set of challenges faced by HCWs in more acute patient care settings like hospital and outpatient clinics, there is a need to further explore the experiences of HCWs within these settings.”

-We appreciate the Butler et al,2021 reference, we included it in our discussion section (not introduction) as we think they are better suited to support our study findings:

Below is the statement added to the discussion section:

“Furthermore, our findings of how lack of transparency and poor communication by HCLs could create mistrust during a crisis was also highlighted by another qualitative study on the experiences of clinicians and clinician leaders in the United States during the COVID-19 pandemic (Butler et al, 2021).”

-We also appreciate the suggestion on the Ohta et al, 2021 reference which we included in our discussion section as it seems better suited sine we elaborate on the study limitation regarding the relatively small number of nursing staff we were able to recruit.

Below is the statement added to the discussion section:

“There is a need for more research that focuses on the unique experiences of nurses given that they spend more time at the bedside with patients compared to other providers and may have experiences unique challenges as a result of their patient facing roles [Ohta et al, 2021].”

-Thank you for the Rosenberg reference. Given that our study is looking specifically at leadership role and support of health care workers during the pandemic, we chose to leave out the Rosenberg et al, 2021 (Journal of pain and symptom management) reference since it focuses more broadly on wellbeing of clinicians (pediatric palliative care physicians) during the pandemic and is not specific to leadership.

Methods

Can the authors clarify their approach to recruitment? 

Thank you for identifying this need for clarification. We added the following statement below in the recruitment section:

“We contacted via email, leaders at various health systems across the state of Connecticut via email including hospital department chairs, division chiefs, residency program directors and nursing leaders. We asked the leaders to forward our recruitment email to their staff. The recruitment email stated that we were inviting health care workers working in Connecticut during the COVID-19 pandemic to participate in an interview to learn about their experiences during the pandemic. All potential participants were further screened for eligibility via a short email survey (see screener email). Individuals were eligible to participate if they were health care workers including resident and attending physicians, respiratory therapists, nurses, patient care assistants (PCA) and certified nurse midwives, worked in Connecticut as a health care worker since March 2020 in the inpatient or outpatient setting or both and had in-person encounters with patients during the pandemic.”

How many HCWs responded to the initial inquiry? 

Were the 45 interviewees selected purposively? If so, what guided this sampling? Was recruitment informed by concurrent analysis? Did the analysis reach thematic saturation?

Thank you for your comment. We added the text below under the “study recruitment” section to address recruitment response and selection of participants:

“A total of 98 HCWs who responded to the initial inquiry. Of these, 38 did not complete the screening questions sent via email. Of the 60 individuals who responded and completed the screening questions sent via email, 2 were ineligible to participate. We interviewed 45 individuals out of the 58 eligible individuals. We purposefully sampled these participants to maximize the number of HCWs who had cared for a relatively high number of patients with COVID-19 diagnosis in hospital in-patient settings who were critically ill and to balance the participants with respect to their specialty and role. We performed concurrent analysis of our interview transcripts and stopped further recruitment and interviewing when we reached thematic saturation. Study participants worked in varying roles and specialties across 3 major HCSs in Connecticut (Table 1).”

Can the authors be more specific on which analytic tools from grounded theory methods they draw from? 

We added the following language to clarify our use of inductive and iterative coding techniques that are derived from grounded theory approaches. We also now include citations to Corbin and Strauss’s 4th edition of Basics of Qualitative Research, a seminal textbook on grounded theory that guided our methodological approach. Please see part of the revised “data analysis” section:

“We used inductive techniques derived from grounded theory to develop themes [Strauss and Corbin, 2014]. Interviewers wrote reflexive memos after each interview and met regularly to discuss emergent concepts. Additionally, we used an iterative and multi-stages coding process that draws on grounded theory approaches [Strauss and Corbin, 2014]. First two of the authors (O.A. and S.T.) reviewed the transcripts and performed open coding of 9 transcripts independently to identify emergent concepts. These concepts were refined through ongoing discussion among all 3 authors and subsequent rounds of open-coding to develop a focused codebook. Two of the authors (O.A. and S.T.) then used this codebook to code the remaining 36 transcripts. All three authors met regularly to discuss code application, making small adjustments where necessary.”

Can they elaborate on the step of “consolidating themes”?

We added the following text in the “data analysis” section to describe our process of consolidating themes:

“To consolidate the leadership themes, one of the authors (O.A.), reviewed all excerpts pertaining to HCWs experiences with or perception of HCLs at all levels of leadership and also reviewed full transcripts to contextualize these excerpts. The first author also wrote integrative memos to develop relationships between codes and concepts and discussed these memos with other team members (D.K. and S.T.).”

Can the authors include a sample interview guide and describe how this guide was developed? Specifically, it is not clear how participants were asked about their experiences with leadership and whether they shared the same understanding of leadership as the authors offer in the introduction. Most of the comments seem to relate to administration or institutional leadership rather than less formal leadership roles. Were the participants asked about their own experience as leaders? The perspective of people with leadership roles might offer a more multi-faceted understanding of the entire phenomenon of leadership during the pandemic from multiple stakeholder perspectives. 

Please find sample interview guide in the supporting information section labeled “S1”.

We added the following statement to the “Data collection” section to describe the development and design of our interview guide and the extent to which participants were asked explicitly about leadership:

“All three authors (O.A., S.T. and D.K.) developed the interview guide through a process of team discussion that was informed by one author’s experience (O.A.) working in the hospital as a physician early in the pandemic and our review of emerging popular and academic literature about HCW experiences. The interview questions focused on broad experiences of HCWs during the pandemic such as their clinical experiences, experiences with supervisors and co-workers, resource limitations, as well as their experiences at home and with the non- medical community (see guide). Specifically, with regards to HCLs, we asked participants open-ended questions that directly explored their perception of leadership’s response and support during the pandemic. These questions include 1) Can you tell me a little bit about how your hospital/ department has responded to COVID-related risks? 2) Is there anything else your institution could have done to support you? Given the semi-structured nature of our questions, we left it up to the participant to choose what aspect of hospital, departmental or institutional leadership they wished to focus on and what “support” meant to them. We also further explored the role of HCLs when participants mentioned the role of leadership in other experiences they had such as availability of personal protective equipment (PPE), impact on their scope of work, etc. We collected self- reported demographic data such as age, gender, race/ethnicity, marital status, number of children, role, specialty and number of patients with COVID-19 cared for.”

-Finally, we agree that it would be interesting to include the perspective of leaders in this study. Our analysis did not explicitly explore participant’s own experiences as leaders during the pandemic, although some of the study participants were leaders such as chief residents and attendings. We also did not disclose the administrative or leadership roles of the participants for anonymity and confidentiality purposes. 

Results

Overall, the findings are mostly descriptive and I wonder if these could be developed a bit more deeply to offer more novel insights if the data is sufficiently rich to support this. For example, (Theme 2, para 3) What is it about some situations that HCWs still wouldn’t enter rooms despite having PPE? (Theme 3, para 2) What is it about volunteerism (vs being made to work) that felt important? (Theme 5) What was it about leaders being in-person that HCWs appreciated? 

Thank you for your suggestion. We have provided more elaboration in these areas. Specifically: 

Theme 2 para 3

Please find an added paragraph that further explores perception of safety by HCWs below:

The above quote also suggests that factors beyond availability of PPE contributed to the feeling of safety by HCWs during the early phases of the pandemic. Uncertainty about the infectivity of the disease, mode of transmission and effectiveness of PPE likely contributed to the complexity of how HCWs perceived safety as the following quote from an OBGYN resident illustrates:

“… And then in the very beginning of the pandemic, the unknown was obviously very anxiety provoking. Was the PPE we were wearing good enough?... Could I contract the virus and bring it home to my wife…?” Participant 32

Notably HCWs developed more confidence in the PPE’s ability to be protective as the following quote from a respiratory therapist illustrates:

 “So overall, it was, I think, one of the most stressful periods of my life. Just because you don’t know if you’re going to get this, if you’re going to bring it home. And of course, you’re protecting yourself as much as possible but they didn’t really know what a lot of information in the beginning like you it was transferred. You know, I guess after the first month or so, it kind of started to become a little bit more normal. And then when you see all these therapist and nurses are directly seeing these patients and they’re not getting sick, whatever we’re doing must be right, the right way… Participant 18”

Theme 3, para 2: What is it about volunteerism (vs being made to work) that felt important? 

 We explored the data some more and added the following text to elaborate on a potential reason why some HCWs did not agree with or like the fact that there were being forced to work:

The tension between volunteering to work versus being forced to work was likely nuanced by the uncertainty about COVID-19 disease and risks of being infected during the early days of the pandemic. The impact of this uncertain risk was noted by another attending physician who stated, 

“…You know the argument being made that this is what you signed up for so you should just do it and stop complaining about it. Because some people, they didn’t want to do. Some people were reasonably uncomfortable to work in such settings and I could understand that. But I heard the argument from some peers again and again that, you know, it’s your job, you need to just do it...And I thought those were inappropriate comments because who signed up to care for a disease that we don’t know much about, without good equipment? I don’t think any of us like when we sign up for medical school to sign up for that specifically….” (Participant 25) 

Theme 5) What was it about leaders being in-person that HCWs appreciated? 

We added a line to the quote by participant 9 that we had previously left out that further explains why the physical presence of the leader was important as this line gives insight to why this participant wanted her supervisor around. 

“…because you want to be able to call someone to be like, you know, I need help …” 

Please see the modified entire text below:

One respiratory therapist described how the physical presence of leadership to troubleshoot challenges provided a sense of comfort particularly if it was a familiar face”

“…And those situations just having more bodies around, like having the supervisors, you know, show up on the floor or that really we rely on our charge therapists a lot for those situations and just having that like you, knowing that somebody is there is definitely more comforting than anything because you want to be able to call someone to be like, you know, I need help…” (Participant 9)

To further elaborate on the importance of physical presence of leaders, we revised the text preceeding the quote by participant 34 that elaborates on the negative effects of not having the health care leader being physically present:

“In contrast, the physical absence of a leader could lead to discouragement as well as a perception that the leader was ineffective. For example, one attending physician explained how physical absence prevented their departmental leader from responding to the rapid changes occurring early in the pandemic. They noted, 

“Unfortunately, our department head, [they] stopped really coming into work…So all of [their]meetings are being run from home and [they] didn't really have [their] finger on the pulse anymore as to what was actually taking place in hospital… I felt quite discouraged by that...” (Participant 34)”

Most of the discussion seems to relate to administrative or institutional leadership more than broad conceptualization of leadership suggested by the authors in the introduction. Can the authors comment on this? 

Thank you for your comments. Though we focus on administrative or institutional leadership as you rightly noted, we also include characteristics relevant to more broad areas of clinical leadership in health care. We discuss transparency and honesty which also pertains to section chiefs, department chairs and residency program directors. We discuss effective communication which was crucial for chief residents, residency program directors, unit managers, section chiefs and departmental chairs. We highlight scheduling considerations and the importance of autonomy which also pertains to chief residents as they make the schedule for other residents as well as section chiefs and chairs who had oversight over attending schedules. 

Discussion.

The authors mention in the introduction that theoretical approaches to leadership during emergency conditions exist, but need to be informed by empirical work. Can they comment on how their work relates to the existing work that they reference?

Please see the paragraph added to the discussion section below:

“Results from our study echo findings from a study performed at the beginning of the beginning of the pandemic about the requests from HCWs to their organization to be heard, protected, prepared, supported and cared for [4]. Our study builds up on this prior work by providing empirical evidence of how HCLs achieved or failed to achieve these requests during the first wave of the pandemic in Connecticut.

 Our study also provides empirical evidence that supports theoretical approaches to leadership during emergency situations and crises such as the importance of transparency, physical presence, effective communication, addressing basic needs and provision of support including mental health support [5-9]”

 • Paragraph 6. I am not sure how the findings support multiple levels of leadership. Can the authors elaborate? •

Thank you for identifying this need for clarification. We added the following language in the “discussion” section to address this comment:

“HCWs in our study described the impact that multiple levels of leadership had on them starting from those with direct oversight such as chief residents (who were involved in scheduling, communication and listening forums), attending physicians, unit managers; those with higher order oversight such as residency program directors, section chiefs, departmental chairs and to those with more centralized leadership roles such as hospital chief executive officers.”

 Discussion might be reinforced by the work of Tannenbaum on team dynamics and leadership (eg, Tannenbaum et al. Managing teamwork in the face of pandemic” BMJ Qual Saf, 2021) 

Thank you for this excellent reference. We have included Tannenbaum’s work in the following added text to our “discussion” section:

“In addition, our findings support existing literature on the responsibility of leaders to provide psychological safety to team members by acknowledging mistakes made by leadership and to create a safe space for dissenting views to be expressed by team members [19].”

Reference section updates

We included the following new references in the reference section

1. Sterling MR, Tseng E, Poon A, Cho J, Avgar A, Kern L, Ankuda C, Dell, N. Experiences of Home Health Care Workers During the Coronavirus Disease 2019 Pandemic A Qualitative Analysis. JAMA Internal Medicine. 2020;180(11):1453-1459. Doi:10.1001/jamainternmed.2020.3930

2. Butler CR, Wong SPY, Vig EK, Neely, CS. BMJ Open. Professional Roles and Relationships during the COVID-19 pandemic: a qualitative study among US clinicians. 2021;11 :e047782.doi:10.1136/bmjopen-2020-047782

3. Tannenbaum SI, Traylor AM, Thomas E, Salas E. BMJ Quality Safety. Managing teamwork in the face of pandemic: evidence0based tips. 2020;0:1-5.doi:10.1136/bmjqs-2020-0111447

4. Ohta R, Matsuzaki Y, Itamoch, S. Overcoming the challenge of COVD-19: A grounded theory approach to rural nurse’s experiences, Journal of General Family Medicine. 2021; 22:134-140

---

## [Decision Letter · Decision Letter 1]

1 Sep 2021

How to lead health care workers during unprecedented crises: A qualitative study of the COVID-19 pandemic in Connecticut, USA

PONE-D-21-11313R1

Dear Dr. Adeyemo,

We’re pleased to inform you that your manuscript has been judged scientifically suitable for publication and will be formally accepted for publication once it meets all outstanding technical requirements.

Kind regards,

Tim Luckett

Academic Editor

PLOS ONE

Reviewer's Responses to Questions

**Comments to the Author**

1. If the authors have adequately addressed your comments raised in a previous round of review and you feel that this manuscript is now acceptable for publication, you may indicate that here to bypass the “Comments to the Author” section, enter your conflict of interest statement in the “Confidential to Editor” section, and submit your "Accept" recommendation.

Reviewer #1: All comments have been addressed

2. Is the manuscript technically sound, and do the data support the conclusions?

Reviewer #1: Yes

3. Has the statistical analysis been performed appropriately and rigorously? 

Reviewer #1: Yes

4. Have the authors made all data underlying the findings in their manuscript fully available?

Reviewer #1: No

5. Is the manuscript presented in an intelligible fashion and written in standard English?

Reviewer #1: Yes

6. Review Comments to the Author

Reviewer #1: I appreciate the authors' thorough and thoughtful revision in response to my comments and excellent work. This manuscript will be a valuable addition to the literature in understanding the complex experience of healthcare workers during the COVID19 pandemic.

7. PLOS authors have the option to publish the peer review history of their article (what does this mean?). If published, this will include your full peer review and any attached files.

Reviewer #1: **Yes: **Catherine Butler

---

## [Editor Report · Acceptance letter]

6 Sep 2021

PONE-D-21-11313R1 

How to lead health care workers during unprecedented crises: A qualitative study of the COVID-19 pandemic in Connecticut, USA 

Dear Dr. Adeyemo:

I'm pleased to inform you that your manuscript has been deemed suitable for publication in PLOS ONE. Congratulations! Your manuscript is now with our production department. 

Kind regards, 

on behalf of

Dr. Tim Luckett 

Academic Editor

PLOS ONE